# Adaptive Classification for Prediction Under a Budget

**Feng Nan**
Systems Engineering
Boston University
Boston, MA 02215
fnan@bu.edu

**Venkatesh Saligrama**
Electrical Engineering
Boston University
Boston, MA 02215
srv@bu.edu

## Abstract

We propose a novel adaptive approximation approach for test-time resource-constrained prediction motivated by Mobile, IoT, health, security and other applications, where constraints in the form of computation, communication, latency and feature acquisition costs arise. We learn an adaptive low-cost system by training a gating and prediction model that limits utilization of a high-cost model to hard input instances and gates easy-to-handle input instances to a low-cost model. Our method is based on adaptively approximating the high-cost model in regions where low-cost models suffice for making highly accurate predictions. We pose an empirical loss minimization problem with cost constraints to jointly train gating and prediction models. On a number of benchmark datasets our method outperforms state-of-the-art achieving higher accuracy for the same cost.

## 1 Introduction

Resource costs arise during test-time prediction in a number of machine learning applications. Feature costs in Internet, Healthcare, and Surveillance applications arise due to to feature extraction time [23], and feature/sensor acquisition [19]. In addition to feature acquisition costs, communication and latency costs pose a key challenge in the design of mobile computing, or the Internet-of-Things(IoT) applications, where a large number of sensors/camera/watches/phones (known as edge devices) are connected to a cloud.

*Adaptive System:* Rather than having the edge devices constantly transmit measurements/images to the cloud where a centralized model makes prediction, a more efficient approach is to allow the edge devices make predictions locally [12], whenever possible, saving the high communication cost and reducing latency. Due to the memory, computing and battery constraints, the prediction models on the edge devices are limited to low complexity. Consequently, to maintain high-accuracy, adaptive systems are desirable. Such systems identify easy-to-handle input instances where local edge models suffice, thus limiting the utilization cloud services for only hard instances. We propose to learn an adaptive system by training on fully annotated training data. Our objective is to maintain high accuracy while meeting average resource constraints during prediction-time.

There have been a number of promising approaches that focus on methods for reducing costs while improving overall accuracy [9, 24, 19, 20, 13, 15]. These methods are adaptive in that, at test-time, resources (features, computation etc) are allocated adaptively depending on the difficulty of the input. Many of these methods train models in a top-down manner, namely, attempt to build out the model by selectively adding the most cost-effective features to improve accuracy.

In contrast we propose a novel bottom-up approach. We train adaptive models on annotated training data by selectively identifying parts of the input space for which high accuracy can be maintained at a lower cost. The principle advantage of our method is twofold. First, our approach can be readily applied to cases where it is desirable to reduce costs of an existing high-cost legacy system. Second, training top-down models in case of feature costs leads to fundamental combinatorial issues in multi-

stage search over all feature subsets (see Sec. 2). In contrast, we bypass many of these issues by posing a natural adaptive approximation objective to partition the input space into easy and hard cases.

In particular, when no legacy system is available, our method consists of first learning a high-accuracy model that minimizes the empirical loss regardless of costs. The resulting high prediction-cost model (HPC) can be readily trained using any of the existing methods. For example, this could be a large neural network in the cloud that achieves the state-of-the-art accuracy. Next, we jointly learn a low-cost gating function as well as a low prediction-cost (LPC) model so as to *adaptively approximate* the high-accuracy model by identifying regions of input space where a low-cost gating and LPC model are adequate to achieve high-accuracy. In IoT applications, such low-complexity models can be deployed on the edge devices to perform gating and prediction. At test-time, for each input instance, the gating function decides whether or not the LPC model is adequate for accurate classification. Intuitively, "easy" examples can be correctly classified using only an LPC model while "hard" examples require HPC model. By identifying which of the input instances can be classified accurately with LPCs we bypass the utilization of HPC model, thus reducing average prediction cost. The upper part of Figure 1 is a schematic of our approach, where $x$ is feature vector and $y$ is the predicted label; we aim to learn $g$ and an LPC model to adaptively approximate the HPC. The key observation as depicted in the lower figure is that the probability of correct classification given $x$ for a HPC model is in general a highly complex function with higher values than that of a LPC model. Yet there exists regions of the input space where the LPC has competitive accuracy (as shown to the right of the gating threshold). Sending examples in such regions (according to the gating function) to the LPC results in no loss of prediction accuracy while reducing prediction costs.

The problem would be simpler if our task were to primarily partition the input space into regions where LPC models would suffice. The difficulty is that we must also learn a low-cost gating function capable of identifying in-

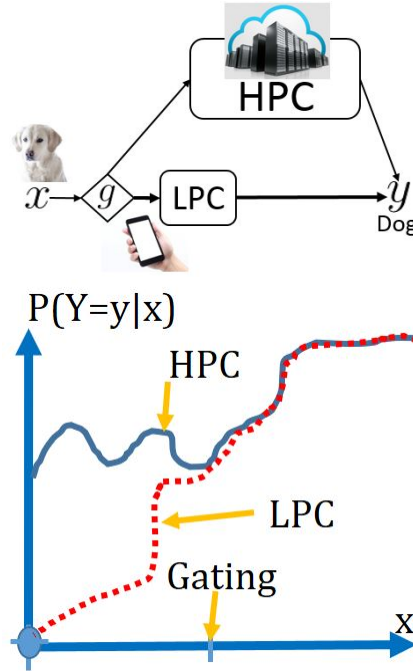

Figure 1: **Upper**: single stage schematic of our approach. We learn low-cost gating $g$ and a LPC model to adaptively approximate a HPC model. **Lower**: Key insight for adaptive approximation. x-axis represents feature space; y-axis represents conditional probability of correct prediction; LPC can match HPC's prediction in the input region corresponding to the right of the gating threshold but performs poorly otherwise. Our goal is to learn a low-cost gating function that attempts to send examples on the right to LPC and the left to HPC.

put instances for which LPC suffices. Since both prediction and gating account for cost, we favor design strategies that lead to shared features and decision architectures between the gating function and the LPC model. We pose the problem as a discriminative empirical risk minimization problem that jointly optimizes for gating and prediction models in terms of a joint margin-based objective function. The resulting objective is separately convex in gating and prediction functions. We propose an alternating minimization scheme that is guaranteed to converge since with appropriate choice of loss-functions (for instance, logistic loss), each optimization step amounts to a probabilistic approximation/projection (I-projection/M-projection) onto a probability space. While our method can be recursively applied in multiple stages to successively approximate the adaptive system obtained in the previous stage, thereby refining accuracy-cost trade-off, we observe that on benchmark datasets even a single stage of our method outperforms state-of-art in accuracy-cost performance.

# 2 Related Work

Learning decision rules to minimize error subject to a budget constraint during prediction-time is an area of active interest[9, 17, 24, 19, 22, 20, 21, 13, 16]. *Pre-trained Models:* In one instantiation

of these methods it is assumed that there exists a collection of prediction models with amortized costs [22, 19, 1] so that a natural ordering of prediction models can be imposed. In other instances, the feature dimension is assumed to be sufficiently low so as to admit an exhaustive enumeration of all the combinatorial possibilities [20, 21]. These methods then learn a policy to choose amongst the ordered prediction models. In contrast we do not impose any of these restrictions. *Top-Down Methods:* For high-dimensional spaces, many existing approaches focus on learning complex adaptive decision functions top-down [9, 24, 13, 21]. Conceptually, during training, top-down methods acquire new features based on their utility value. This requires exploration of partitions of the input space together with different combinatorial low-cost feature subsets that would result in higher accuracy. These methods are based on multi-stage exploration leading to combinatorially hard problems. Different novel relaxations and greedy heuristics have been developed in this context. *Bottom-up Methods:* Our work is somewhat related to [16], who propose to prune a fully trained random forests (RF) to reduce costs. Nevertheless, in contrast to our adaptive system, their perspective is to compress the original model and utilize the pruned forest as a stand-alone model for test-time prediction. Furthermore, their method is specifically tailored to random forests.

Another set of related work includes classifier cascade [5] and decision DAG [3], both of which aim to re-weight/re-order a set of pre-trained base learners to reduce prediction budget. Our method, on the other hand, only requires to pre-train a high-accuracy model and jointly learns the low-cost models to approximate it; therefore ours can be viewed as complementary to the existing work. The teacher-student framework [14] is also related to our bottom-up approach; a low-cost student model learns to approximate the teacher model so as to meet test-time budget. However, the goal there is to learn a better stand-alone student model. In contrast, we make use of both the low-cost (student) and high-accuracy (teacher) model during prediction via a gating function, which learns the limitation of the low-cost (student) model and consult the high-accuracy (teacher) model if necessary, thereby avoiding accuracy loss. Our composite system is also related to HME [10], which learns the composite system based on max-likelihood estimation of models. A major difference is that HME does not address budget constraints. A fundamental aspect of budget constraints is the resulting asymmetry, whereby, we start with an HPC model and sequentially approximate with LPCs. This asymmetry leads us to propose a bottom-up strategy where the high-accuracy predictor can be separately estimated and is critical to posing a direct empirical loss minimization problem.

## 3 Problem Setup

We consider the standard learning scenario of resource constrained prediction with feature costs. A training sample $S = \{(x^{(i)}, y^{(i)}) : i = 1, \ldots, N\}$ is generated i.i.d. from an unknown distribution, where $x^{(i)} \in \Re^K$ is the feature vector with an acquisition cost $c_\alpha \geq 0$ assigned to each of the features $\alpha = 1, \ldots, K$ and $y^{(i)}$ is the label for the $i^{\text{th}}$ example. In the case of multi-class classification $y \in \{1, \ldots, M\}$, where $M$ is the number of classes. Let us consider a single stage of our training method in order to formalize our setup. The model, $f_0$, is a high prediction-cost (HPC) model, which is either a priori known, or which we train to high-accuracy regardless of cost considerations. We would like to learn an alternative low prediction-cost (LPC) model $f_1$. Given an example $x$, at test-time, we have the option of selecting which model, $f_0$ or $f_1$, to utilize to make a prediction. The accuracy of a prediction model $f_z$ is modeled by a loss function $\ell(f_z(x), y), z \in \{0, 1\}$. We exclusively employ the logistic loss function in binary classification: $\ell(f_z(x), y) = \log(1 + \exp(-yf_z(x)))$, although our framework allows other loss models. For a given $x$, we assume that once it pays the cost to acquire a feature, its value can be efficiently cached; its subsequent use does not incur additional cost. Thus, the cost of utilizing a particular prediction model, denoted by $c(f_z, x)$, is computed as the sum of the acquisition cost of *unique* features required by $f_z$.

**Oracle Gating:** Consider a general gating likelihood function $q(z|x)$ with $z \in \{0, 1\}$, that outputs the likelihood of sending the input $x$ to a prediction model, $f_z$. The overall empirical loss is:

$$\mathbb{E}_{S_n}\mathbb{E}_{q(z|x)}[\ell(f_z(x), y)] = \mathbb{E}_{S_n}[\ell(f_0(x), y)] + \mathbb{E}_{S_n}\big[q(1|x)\underbrace{(\ell(f_1(x), y) - \ell(f_0(x), y))}_{Excess\ Loss}\big]$$

The first term only depends on $f_0$, and from our perspective a constant. Similar to average loss we can write the average cost as (assuming gating cost is negligible for now):

$$\mathbb{E}_{S_n}\mathbb{E}_{q(z|x)}[c(f_z, x)] = \mathbb{E}_{S_n}[c(f_0, x)] - \mathbb{E}_{S_n}[q(1|x)\underbrace{(c(f_0, x) - c(f_1, x))}_{\textit{Cost Reduction}}],$$

where the first term is again constant. We can characterize the optimal gating function (see [19]) that minimizes the overall average loss subject to average cost constraint:

$$\overbrace{\ell(f_1, x) - \ell(f_0, x)}^{\textit{Excess loss}} \underset{q(1|x)=1}{\overset{q(1|x)=0}{\gtrless}} \eta \overbrace{(c(f_0, x) - c(f_1, x))}^{\textit{Cost reduction}}$$

for a suitable choice $\eta \in \mathbb{R}$. This characterization encodes the important principle that if the marginal cost reduction is smaller than the excess loss, we opt for the HPC model. Nevertheless, this characterization is generally infeasible. Note that the LHS depends on knowing how well HPC performs on the input instance. Since this information is unavailable, this target can be unreachable with low-cost gating.

**Gating Approximation:** Rather than directly enforcing a low-cost structure on $q$, we decouple the constraint and introduce a parameterized family of gating functions $g \in \mathcal{G}$ that attempts to mimic (or approximate) $q$. To ensure such approximation, we can minimize some distance measure $D(q(\cdot|x), g(x))$. A natural choice for an approximation metric is the Kullback-Leibler (KL) divergence although other choices are possible. The KL divergence between $q$ and $g$ is given by $D_{KL}(q(\cdot|x)\|g(x)) = \sum_z q(z|x)\log(q(z|x)/\sigma(\mathrm{sgn}(0.5 - z)g(x)))$, where $\sigma(s) = 1/(1 + e^{-s})$ is the sigmoid function. Besides KL divergence, we have also proposed another symmetrized metric fitting $g$ directly to the log odds ratio of $q$. See Suppl. Material for details.

**Budget Constraint:** With the gating function $g$, the cost of predicting $x$ depends on whether the example is sent to $f_0$ or $f_1$. Let $c(f_0, g, x)$ denote the feature cost of passing $x$ to $f_0$ through $g$. As discussed, this is equal to the sum of the acquisition cost of unique features required by $f_0$ and $g$ for $x$. Similarly $c(f_1, g, x)$ denotes the cost if $x$ is sent to $f_1$ through $g$. In many cases the cost $c(f_z, g, x)$ is independent of the example $x$ and depends primarily on the model being used. This is true for linear models where each $x$ must be processed through the same collection of features. For these cases $c(f_z, g, x) \triangleq c(f_z, g)$. The total budget simplifies to: $\mathbb{E}_{S_n}[q(0|x)]c(f_0, g) + (1 - \mathbb{E}_{S_n}[q(0|x)])c(f_1, g) = c(f_1, g) + \mathbb{E}_{S_n}[q(0|x)](c(f_0, g) - c(f_1, g))$. The budget thus depends on 3 quantities: $\mathbb{E}_{S_n}[q(0|x)]$, $c(f_1, g)$ and $c(f_0, g)$. Often $f_0$ is a high-cost model that requires most, if not all, of features so $c(f_0, g)$ can be considered a large constant.

Thus, to meet the budget constraint, we would like to have (a) low-cost $g$ and $f_1$ (small $c(f_1, g)$); and (b) small fraction of examples being sent to the high-accuracy model (small $\mathbb{E}_{S_n}[q(0|x)]$). We can therefore split the budget constraint into two separate objectives: (a) ensure low-cost through penalty $\Omega(f_1, g) = \gamma \sum_\alpha c_\alpha \|V_\alpha + W_\alpha\|_0$, where $\gamma$ is a tradeoff parameter and the indicator variables $V_\alpha, W_\alpha \in \{0, 1\}$ denote whether or not the feature $\alpha$ is required by $f_1$ and $g$, respectively. Depending on the model parameterization, we can approximate $\Omega(f_1, g)$ using a group-sparse norm or in a stage-wise manner as we will see in Algorithms 1 and 2. (b) Ensure only $\mathrm{P}_{\text{full}}$ fraction of examples are sent to $f_0$ via the constraint $\mathbb{E}_{S_n}[q(0|x)] \leq \mathrm{P}_{\text{full}}$.

**Putting Together:** We are now ready to pose our general optimization problem:

$$\min_{f_1 \in \mathcal{F}, g \in \mathcal{G}, q} \mathbb{E}_{S_n} \overbrace{\sum_z [q(z|x)\ell(f_z(x), y)]}^{\textit{Losses}} + \overbrace{D(q(\cdot|x), g(x))}^{\textit{Gating Approx}} + \overbrace{\Omega(f_1, g)}^{\textit{Costs}} \tag{OPT}$$

$$\text{subject to: } \mathbb{E}_{S_n}[q(0|x)] \leq \mathrm{P}_{\text{full}}. \; \textit{(Fraction to } f_0\textit{)}$$

The objective function penalizes excess loss and ensures through the second term that this excess loss can be enforced through admissible gating functions. The third term penalizes the feature cost usage of $f_1$ and $g$. The budget constraint limits the fraction of examples sent to the costly model $f_0$.

*Remark 1*: Directly parameterizing $q$ leads to non-convexity. Average loss is $q$-weighted sum of losses from HPC and LPC; while the space of probability distributions is convex, a finite-dimensional parameterization is generally non-convex (e.g. sigmoid). What we have done is to keep $q$ in non-parametric form to avoid non-convexity and only parameterize $g$, connecting both via

a KL term. Thus, (OPT) is now convex with respect to the $f_1$ and $g$ for a fixed $q$. It is again convex in $q$ for a fixed $f_1$ and $g$. Otherwise it would introduce non-convexity as in prior work. For instance, in [5] a non-convex problem is solved in each inner loop iteration (line 7 of their Algorithm 1).

*Remark 2*: We presented the case for a single stage approximation system. However, it is straightforward to recursively continue this process. We can then view the composite system $f_0 \triangleq (g, f_1, f_0)$ as a black-box predictor and train a new pair of gating and prediction models to approximate the composite system.

*Remark 3*: To limit the scope of our paper, we focus on reducing feature acquisition cost during prediction as it is a more challenging (combinatorial) problem. However, other prediction-time costs such as computation cost can be encoded in the choice of functional classes $\mathcal{F}$ and $\mathcal{G}$ in (OPT).

*Surrogate Upper Bound of Composite System*: We can get better insight for the first two terms of the objective in (OPT) if we view $z \in \{0, 1\}$ as a latent variable and consider the composite system $\Pr(y|x) = \sum_z \Pr(z|x; g) \Pr(y|x, f_z)$. A standard application of Jensen's inequality reveals that, $-\log(\Pr(y|x)) \le \mathbb{E}_{q(z|x)}\ell(f_z(x), y) + D_{KL}(q(z|x)\| \Pr(z|x; g))$. Therefore, the conditional-entropy of the composite system is bounded by the expected value of our loss function (we overload notation and represent random-variables in lower-case format):

$$H(y \mid x) \triangleq \mathbb{E}[-\log(\Pr(y|x))] \le \mathbb{E}_{x \times y}[\mathbb{E}_{q(z|x)}\ell(f_z(x), y) + D_{KL}(q(z|x)\| \Pr(z|x; g))].$$

This implies that the first two terms of our objective attempt to bound the loss of the composite system; the third term in the objective together with the constraint serve to enforce budget limits on the composite system.

*Group Sparsity:* Since the cost for feature re-use is zero we encourage feature re-use among gating and prediction models. So the fundamental question here is: *How to choose a common, sparse (low-cost) subset of features on which both $g$ and $f_1$ operate, such that $g$ can effective gate examples between $f_1$ and $f_0$ for accurate prediction?* This is a hard combinatorial problem. The main contribution of our paper is to address it using the general optimization framework of (OPT).

## 4 Algorithms

To be concrete, we instantiate our general framework (OPT) into two algorithms via different parameterizations of $g, f_1$: ADAPT-LIN for the linear class and ADAPT-GBRT for the non-parametric class. Both of them use the KL-divergence as distance measure. We also provide a third algorithm ADAPT-LSTSQ that uses the symmetrized distance in the Suppl. Material. All of the algorithms perform alternating minimization of (OPT) over $q, g, f_1$. Note that convergence of alternating minimization follows as in [8]. Common to all of our algorithms, we use two parameters to control cost: $\mathrm{P_{full}}$ and $\gamma$. In practice they are swept to generate various cost-accuracy tradeoffs and we choose the best one satisfying the budget $B$ using validation data.

**ADAPT-LIN:** Let $g(x) = g^T x$ and $f_1(x) = f_1^T x$ be linear classifiers. A feature is used if the corresponding component is non-zero: $V_\alpha = 1$ if $f_{1,\alpha} \ne 0$, and $W_\alpha = 1$ if $g_\alpha \ne 0$. The minimization for $q$ solves the following problem:

$$\min_q \quad \frac{1}{N}\sum_{i=1}^N [(1-q_i)A_i + q_i B_i - H(q_i)]$$
$$\text{s.t.} \quad \frac{1}{N}\sum_{i=1}^N q_i \le \mathrm{P_{full}}, \tag{OPT1}$$

where we have used shorthand notations $q_i = q(z = 0|x^{(i)})$, $H(q_i) = -q_i \log(q_i) - (1 - q_i)\log(1 - q_i)$, $A_i = \log(1 + e^{-y^{(i)}f_1^T x^{(i)}}) +$

---

**Algorithm 1** ADAPT-LIN

**Input:** $(x^{(i)}, y^{(i)}), \mathrm{P_{full}}, \gamma$
Train $f_0$. Initialize $g, f_1$.
**repeat**
    Solve (OPT1) for $q$ given $g, f_1$.
    Solve (OPT2) for $g, f_1$ given $q$.
**until** convergence

---

**Algorithm 2** ADAPT-GBRT

**Input:** $(x^{(i)}, y^{(i)}), \mathrm{P_{full}}, \gamma$
Train $f_0$. Initialize $g, f_1$.
**repeat**
    Solve (OPT1) for $q$ given $g, f_1$.
    **for** $t = 1$ **to** $T$ **do**
        Find $f_1^t$ using CART to minimize (1).
        $f_1 = f_1 + f_1^t$.
        For each feature $\alpha$ used, set $u_\alpha = 0$.
        Find $g^t$ using CART to minimize (2).
        $g = g + g^t$.
        For each feature $\alpha$ used, set $u_\alpha = 0$.
    **end for**
**until** convergence

$\log(1 + e^{g^T x^{(i)}})$ and $B_i = -\log p(y^{(i)}|z^{(i)} = 0; f_0) + \log(1 + e^{-g^T x^{(i)}})$. This optimization has a closed form solution: $q_i = 1/(1 + e^{B_i - A_i + \beta})$ for some non-negative constant $\beta$ such that the constraint is satisfied. This optimization is also known as I-Projection in information geometry because of the entropy term [8]. Having optimized $q$, we hold it constant and minimize with respect to $g, f_1$ by solving the problem (OPT2), where we have relaxed the non-convex cost $\sum_\alpha c_\alpha \|V_\alpha + W_\alpha\|_0$ into a $L_{2,1}$ norm for group sparsity and a tradeoff parameter $\gamma$ to make sure the feature budget is satisfied. Once we solve for $g, f_1$, we can hold them constant and minimize with respect to $q$ again. ADAPT-LIN is summarized in Algorithm 1.

$$\min_{g,f_1} \frac{1}{N} \sum_{i=1}^{N} \left[ (1 - q_i) \left( \log(1 + e^{-y^{(i)} f_1^T x^{(i)}}) + \log(1 + e^{g^T x^{(i)}}) \right) + q_i \log(1 + e^{-g^T x^{(i)}}) \right] + \gamma \sum_\alpha \sqrt{g_\alpha^2 + f_{1,\alpha}^2}. \tag{OPT2}$$

**ADAPT-GBRT:** We can also consider the non-parametric family of classifiers such as gradient boosted trees [7]: $g(x) = \sum_{t=1}^{T} g^t(x)$ and $f_1(x) = \sum_{t=1}^{T} f_1^t(x)$, where $g^t$ and $f_1^t$ are limited-depth regression trees. Since the trees are limited to low depth, we assume that the feature utility of each tree is example-independent: $V_{\alpha,t}(x) \cong V_{\alpha,t}, W_{\alpha,t}(x) \cong W_{\alpha,t}, \forall x$. $V_{\alpha,t} = 1$ if feature $\alpha$ appears in $f_1^t$, otherwise $V_{\alpha,t} = 0$, similarly for $W_{\alpha,t}$. The optimization over $q$ still solves (OPT1). We modify $A_i = \log(1 + e^{-y^{(i)} f_1(x^{(i)})}) + \log(1 + e^{g(x^{(i)})})$ and $B_i = -\log p(y^{(i)}|z^{(i)} = 0; f_0) + \log(1 + e^{-g(x^{(i)})})$. Next, to minimize over $g, f_1$, denote loss:

$$\ell(f_1, g) = \frac{1}{N} \sum_{i=1}^{N} \left[ (1 - q_i) \cdot \left( \log(1 + e^{-y^{(i)} f_1(x^{(i)})}) + \log(1 + e^{g(x^{(i)})}) \right) + q_i \log(1 + e^{-g(x^{(i)})}) \right],$$

which is essentially the same as the first part of the objective in (OPT2). Thus, we need to minimize $\ell(f_1, g) + \Omega(f_1, g)$ with respect to $f_1$ and $g$. Since both $f_1$ and $g$ are gradient boosted trees, we naturally adopt a stage-wise approximation for the objective. In particular, we define an impurity function which on the one hand approximates the negative gradient of $\ell(f_1, g)$ with the squared loss, and on the other hand penalizes the initial acquisition of features by their cost $c_\alpha$. To capture the initial acquisition penalty, we let $u_\alpha \in \{0, 1\}$ indicates if feature $\alpha$ has already been used in previous trees ($u_\alpha = 0$), or not ($u_\alpha = 1$). $u_\alpha$ is updated after adding each tree. Thus we arrive at the following impurity for $f_1$ and $g$, respectively:

$$\frac{1}{2} \sum_{i=1}^{N} (-\frac{\partial \ell(f_1, g)}{\partial f_1(x^{(i)})} - f_1^t(x^{(i)}))^2 + \gamma \sum_\alpha u_\alpha c_\alpha V_{\alpha,t}, \tag{1}$$

$$\frac{1}{2} \sum_{i=1}^{N} (-\frac{\partial \ell(f_1, g)}{\partial g(x^{(i)})} - g^t(x^{(i)}))^2 + \gamma \sum_\alpha u_\alpha c_\alpha W_{\alpha,t}. \tag{2}$$

Minimizing such impurity functions balances the need to minimize loss and re-using the already acquired features. Classification and Regression Tree (CART) [2] can be used to construct decision trees with such an impurity function. ADAPT-GBRT is summarized in Algorithm 2. Note that a similar impurity is used in GREEDYMISER [24]. Interestingly, if P$_\text{full}$ is set to 0, all the examples are forced to $f_1$, then ADAPT-GBRT exactly recovers the GREEDYMISER. In this sense, GREEDYMISER is a special case of our algorithm. As we will see in the next section, thanks to the bottom-up approach, ADAPT-GBRT benefits from high-accuracy initialization and is able to perform accuracy-cost tradeoff in accuracy levels beyond what is possible for GREEDYMISER.

## 5  Experiments

**BASELINE ALGORITHMS:** We consider the following simple L1 baseline approach for learning $f_1$ and $g$: first perform a L1-regularized logistic regression on all data to identify a relevant, sparse subset of features; then learn $f_1$ using training data restricted to the identified feature(s); finally, learn $g$ based on the correctness of $f_1$ predictions as pseudo labels (i.e. assign pseudo label 1 to example $x$ if $f_1(x)$ agrees with the true label $y$ and 0 otherwise). We also compare with two state-of-the-art feature-budgeted algorithms: GREEDYMISER[24] - a top-down method that builds out an

ensemble of gradient boosted trees with feature cost budget; and BUDGETPRUNE[16] - a bottom-up method that prunes a random forest with feature cost budget. A number of other methods such as ASTC [13] and CSTC [23] are omitted as they have been shown to under-perform GREEDYMISER on the same set of datasets [15]. Detailed experiment setups can be found in the Suppl. Material.

We first visualize/verify the adaptive approximation ability of ADAPT-LIN and ADAPT-GBRT on the Synthetic-1 dataset without feature costs. Next, we illustrate the key difference between ADAPT-LIN and the L1 baseline approach on the Synthetic-2 as well as the Letters datasets. Finally, we compare ADAPT-GBRT with state-of-the-art methods on several resource constraint benchmark datasets.

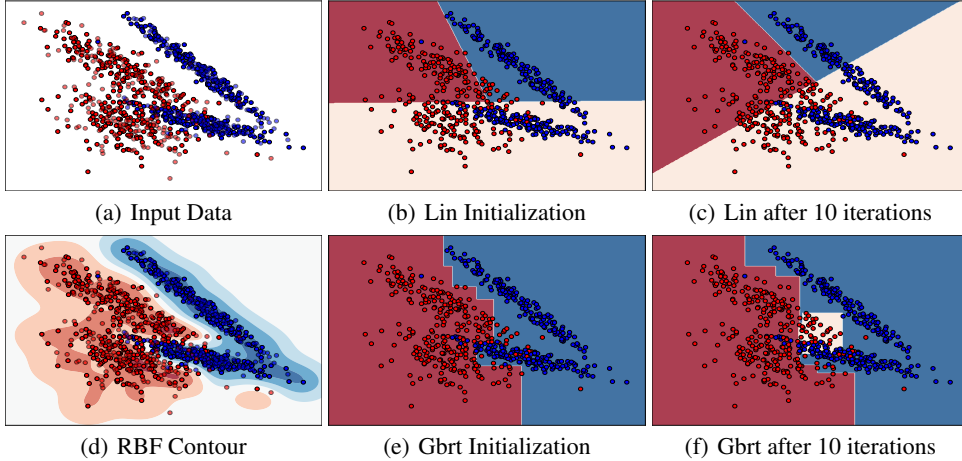

(a) Input Data      (b) Lin Initialization      (c) Lin after 10 iterations

(d) RBF Contour      (e) Gbrt Initialization      (f) Gbrt after 10 iterations

Figure 2: Synthetic-1 experiment without feature cost. (a): input data. (d): decision contour of RBF-SVM as $f_0$. (b) and (c): decision boundaries of linear $g$ and $f_1$ at initialization and after 10 iterations of ADAPT-LIN. (e) and (f): decision boundaries of boosted tree $g$ and $f_1$ at initialization and after 10 iterations of ADAPT-GBRT. Examples in the beige areas are sent to $f_0$ by the $g$.

**POWER OF ADAPTATION:** We construct a 2D binary classification dataset (Synthetic-1) as shown in (a) of Figure 2. We learn an RBF-SVM as the high-accuracy classifier $f_0$ as in (d). To better visualize the adaptive approximation process in 2D, we turn off the feature costs (i.e. set $\Omega(f_1, g)$ to 0 in (OPT)) and run ADAPT-LIN and ADAPT-GBRT.

The initializations of $g$ and $f_1$ in (b) results in wrong predictions for many red points in the blue region. After 10 iterations of ADAPT-LIN, $f_1$ adapts much better to the local region assigned by $g$ while $g$ sends about 60% ($P_{\text{full}}$) of examples to $f_0$. Similarly, the initialization in (e) results in wrong predictions in the blue region. ADAPT-GBRT is able to identify the ambiguous region in the center and send those examples to $f_0$ via $g$. Both of our algorithms maintain the same level of prediction accuracy as $f_0$ yet are able to classify large fractions of examples via much simpler models.

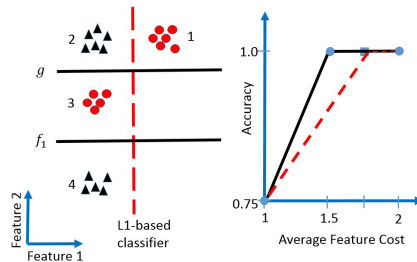

**POWER OF JOINT OPTIMIZATION:** We return to the problem of prediction under feature budget constrains. We illustrate why a simple L1 baseline approach for learning $f_1$ and $g$ would not work using a 2D dataset (Synthetic-2) as shown in Figure 3 (left). The data points are distributed in four clusters, with black triangles and red circles representing two class labels. Let both feature 1 and 2 carry unit acquisition cost. A complex classifier $f_0$ that acquires both features can achieve full accuracy

Figure 3: A 2-D synthetic example for adaptive feature acquisition. On the left: data distributed in four clusters. The two features correspond to x and y co-ordinates, respectively. On the right: accuracy-cost tradeoff curves. Our al-gorithm can recover the optimal adap-tive system whereas a L1-based ap-proach cannot.

at the cost of 2. In our synthetic example, clusters 1 and 2 are given more data points so that the L1-regularized logistic regression would produce the vertical red dashed line, separating cluster 1 from the others. So feature 1 is acquired for both $g$ and $f_1$. The best such an adaptive system can

do is to send cluster 1 to $f_1$ and the other three clusters to the complex classifier $f_0$, incurring an average cost of 1.75, which is sub-optimal. ADAPT-LIN, on the other hand, optimizing between $q, g, f_1$ in an alternating manner, is able to recover the horizontal lines in Figure 3 (left) for $g$ and $f_1$. $g$ sends the first two clusters to the full classifier and the last two clusters to $f_1$. $f_1$ correctly classifies clusters 3 and 4. So all of the examples are correctly classified by the adaptive system; yet only feature 2 needs to be acquired for cluster 3 and 4 so the overall average feature cost is 1.5, as shown by the solid curve in the accuracy-cost tradeoff plot on the right of Figure 3. This example shows that the L1 baseline approach is sub-optimal as it doesnot optimize the selection of feature subsets *jointly* for $g$ and $f_1$.

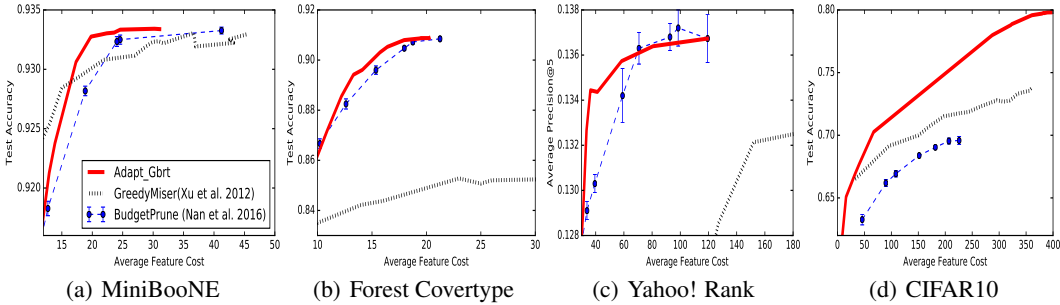

(a) MiniBooNE     (b) Forest Covertype     (c) Yahoo! Rank     (d) CIFAR10

Figure 4: Comparison of ADAPT-GBRT against GREEDYMISER and BUDGETPRUNE on four benchmark datasets. RF is used as $f_0$ for ADAPT-GBRT in (a-c) while an RBF-SVM is used as $f_0$ in (d). ADAPT-GBRT achieves better accuracy-cost tradeoff than other methods. The gap is significant in (b) (c) and (d). Note the accuracy of GREEDYMISER in (b) never exceeds 0.86 and its precision in (c) slowly rises to 0.138 at cost of 658. We limit the cost range for a clearer comparison.

**REAL DATASETS:** We test various aspects of our algorithms and compare with state-of-the-art feature-budgeted algorithms on five real world benchmark datasets: Letters, Mini-BooNE Particle Identification, Forest Covertype datasets from the UCI repository [6], CIFAR-10 [11] and Yahoo! Learning to

Table 1: Dataset Statistics

| Dataset | #Train | #Validation | #Test | #Features | Feature Costs |
|---|---|---|---|---|---|
| Letters | 12000 | 4000 | 4000 | 16 | Uniform |
| MiniBooNE | 45523 | 19510 | 65031 | 50 | Uniform |
| Forest | 36603 | 15688 | 58101 | 54 | Uniform |
| CIFAR10 | 19761 | 8468 | 10000 | 400 | Uniform |
| Yahoo! | 141397 | 146769 | 184968 | 519 | CPU units |

Rank[4]. Yahoo! is a ranking dataset where each example is associated with features of a query-document pair together with the relevance rank of the document to the query. There are 519 such features in total; each is associated with an acquisition cost in the set {1,5,20,50,100,150,200}, which represents the units of CPU time required to extract the feature and is provided by a Yahoo! employee. The labels are binarized into relevant or not relevant. The task is to learn a model that takes a new query and its associated documents and produce a relevance ranking so that the relevant documents come on top, and to do this using as little feature cost as possible. The performance metric is Average Precision @ 5 following [16]. The other datasets have unknown feature costs so we assign costs to be 1 for all features; the aim is to show ADAPT-GBRT successfully selects *sparse* subset of "usefull" features for $f_1$ and $g$. We summarize the statistics of these datasets in Table 1. Next, we highlight the key insights from the real dataset experiments.

**Generality of Approximation:** Our framework allows approximation of powerful classifiers such as RBF-SVM and Random Forests as shown in Figure 5 as red and black curves, respectively. In particular, ADAPT-GBRT can well maintain high accuracy while reducing cost. This is a key advantage for our algorithms because we can choose to approximate the $f_0$ that achieves the best accuracy. **ADAPT-LIN Vs L1:** Figure 5 shows that ADAPT-LIN outperforms L1 baseline method on real dataset as well. Again, this confirms the intuition we have in the Synthetic-2 example as ADAPT-LIN is able to iteratively select the common subset of features jointly for $g$ and $f_1$. **ADAPT-GBRT Vs ADAPT-LIN:** ADAPT-GBRT leads to significantly better performance than ADAPT-LIN in approximating both RBF-SVM and RF as shown in Figure 5. This is expected as the non-parametric non-linear classifiers are much more powerful than linear ones.

**ADAPT-GBRT Vs BUDGETPRUNE:** Both are bottom-up approaches that benefit from good initializations. In (a), (b) and (c) of Figure 4 we let $f_0$ in ADAPT-GBRT be the same RF that BUDGET-PRUNE starts with. ADAPT-GBRT is able to maintain high accuracy longer as the budget decreases.

Thus, ADAPT-GBRT improves state-of-the-art bottom-up method. Notice in (c) of Figure 4 around the cost of 100, BUDGETPRUNE has a spike in precision. We believe this is because the initial pruning improved the generalization performance of RF.

But in the cost region of 40-80, ADAPT-GBRT maintains much better accuracy than BUDGET-PRUNE. Furthermore, ADAPT-GBRT has the freedom to approximate the best $f_0$ given the problem. So in (d) of Figure 4 we see that with $f_0$ being RBF-SVM, ADAPT-GBRT can achieve much higher accuracy than BUDGETPRUNE.

**ADAPT-GBRT VS GREEDYMISER:** ADAPT-GBRT outperforms GREEDYMISER on all the datasets. The gaps in Figure 5, (b) (c) and (d) of Figure 4 are especially significant.

**Significant Cost Reduction:** Without sacrificing top accuracies (within 1%), ADAPT-GBRT reduces average feature costs during test-time by around 63%, 32%, 58%, 12% and 31% on MiniBooNE, Forest, Yahoo, Cifar10 and Letters datasets, respectively.

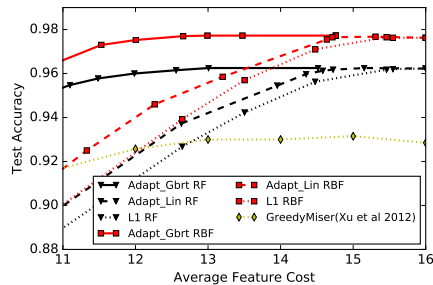

Figure 5: Compare the L1 baseline approach, ADAPT-LIN and ADAPT-GBRT based on RBF-SVM and RF as $f_0$'s on the Letters dataset.

## 6  Conclusions

We presented an adaptive approximation approach to account for prediction costs that arise in various applications. At test-time our method uses a gating function to identify a prediction model among a collection of models that is adapted to the input. The overall goal is to reduce costs without sacrificing accuracy. We learn gating and prediction models by means of a bottom-up strategy that trains low prediction-cost models to approximate high prediction-cost models in regions where low-cost models suffice. On a number of benchmark datasets our method leads to an average of 40% cost reduction without sacrificing test accuracy (within 1%). It outperforms state-of-the-art top-down and bottom-up budgeted learning algorithms, with a significant margin in several cases.

**Acknowledgments**

Feng Nan would like to thank Dr Ofer Dekel for ideas and discussions on resource constrained machine learning during an internship in Microsoft Research in summer 2016. Familiarity and intuition gained during the internship contributed to the motivation and formulation in this paper. We also thank Dr Joseph Wang and Tolga Bolukbasi for discussions and helps in experiments. This material is based upon work supported in part by NSF Grants CCF: 1320566, CNS: 1330008, CCF: 1527618, DHS 2013-ST-061-ED0001, NGA Grant HM1582-09-1-0037 and ONR Grant N00014-13-C-0288.

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
