[Supplementary Material]

# Supplementary Material: Adaptive Classification for Prediction Under a Budget

May 19, 2017

## 0.1 ADAPT-LSTSQ

**Other Symmetrized metrics:** KL divergence is not symmetric and leads to widely different properties in terms of approximation. We also consider a symmetrized metric:

$$D(r(z), s(z)) = \left(\log \frac{r(0)}{r(1)} - \log \frac{s(0)}{s(1)}\right)^2$$

This metric can be viewed intuitively as a regression of $g(x) = \log(\Pr(1|g; x)/\Pr(0|g; x))$ against the observed log odds ratio of $q(z|x)$.

The main advantage of using KL is that optimizing w.r.t. $q$ can be solved in closed form. The disadvantage we observe is that in some cases, the loss for minimizing w.r.t. $g$, which is a weighted sum of log-losses of opposing directions, becomes quite flat and difficult to optimize especially for linear gating functions. The symmetrized measure, on the other hand, makes the optimization w.r.t. $g$ better conditioned as the gating function $g$ fits directly to the log odds ratio of $q$. However, the disadvantage of using the symmetrized measure is that optimizing w.r.t. $q$ no longer has closed form solution; furthermore, it is even non-convex. We offer an ADMM approach for $q$ optimization.

We still follow an alternating minimization approach. To keep the presentation simply, we assume $g, f_1$ to be linear classifiers and there is no feature costs involved. To minimize over $q$, we must solve

$$\begin{aligned} \min_{q_i \in [0,1]} \quad & \frac{1}{N} \sum_{i=1}^{N} \left[ (1 - q_i)A_i + (\log \frac{q_i}{1-q_i} - g(x^{(i)}))^2 \right] \\ \text{s.t.} \quad & \frac{1}{N} \sum_{i=1}^{N} q_i \leq P_{\text{full}}, \end{aligned} \tag{OPT5}$$

where $q_i = q(z = 0|x^{(i)})$, $A_i = \log(1 + e^{-y^{(i)} f_1^T x^{(i)}}) + \log p(y^{(i)}|z^{(i)} = 1; f_0)$. Unlike (OPT3), this optimization problem no longer has a closed-form solution. Fortunately, the $q_i$'s in the objective are decoupled and there is only one coupling constraint. We can solve this problem using an ADMM approach [1]. To optimize over $g$, we simply need to solve a linear least squares problem:

$$\min_g \frac{1}{N} \sum_{i=1}^{N} (\log \frac{q_i}{1-q_i} - g^T(x^{(i)}))^2. \tag{OPT6}$$

To optimize over $f_1$, we solve a weighted logistic regression problem:

$$\min_{f_1} \frac{1}{N} \sum_{i=1}^{N} (1 - q_i) \log(1 + e^{-y^{(i)} f_1^T x^{(i)}}). \tag{OPT7}$$

We shall call the above algorithm ADAPT-LSTSQ, summarized in Algorithm 1. However, on a number of datasets, we found that ADAPT-LSTSQ is comparable to ADAPT-GBRT thus we did not include it in the main plots.

## 0.2 Experimental Details

We provide detailed parameter settings and steps for our experiments here.

---
**Algorithm 1** ADAPT-LSTSQ
---
   **Input:** $(x^{(i)}, y^{(i)}), B$
   Train a full accuracy model $f_0$.
   Initialize $g, f_1$.
   **repeat**
      Solve (OPT5) for $q$ given $g, f_1$.
      Solve (OPT6) for $g$ given $q$.
      Solve (OPT7) for $f_1$ given $q$.
   **until** convergence
---

## 0.3  Synthetic-1 Experiment

We generate the data in Python using the following command:

```
X, y = make_classification(n_samples=1000, flip_y=0.01, n_features=2,
n_redundant=0, n_informative=2,random_state=17, n_clusters_per_class=2)
```

For ADAPT-GBRT we used 5 depth-2 trees for $g$ and $f_1$.

## 0.4  Synthetic-2 Experiment:

We generate 4 clusters on a 2D plane with centers: (1,1), (-1,1), (-1,-1), (-1, -3) and Gaussian noise with standard deviation of 0.01. The first two clusters have 20 examples each and the last two clusters have 15 examples each. We sweep the regularization parameter of L1-regularized logistic regression and recover feature 1 as the sparse subset, which leads to sub-optimal adaptive system. On the other hand, we can easily train a RBF SVM classifier to correctly classify all clusters and we use it as $f_0$. If we initialize $g$ and $f_1$ with Gaussian distribution centered around 0, ADAPT-LIN with can often recover feature 2 as the sparse subset and learn the correct $g$ and $f_1$. Or, we could initialize $g = (1, 1)$ and $f_1 = (1, 1)$ then ADAPT-LIN can recover the optimal solution.

## 0.5  Letters Dataset [4]

This letters recognition dataset contains 20000 examples with 16 features, each of which is assigned unit cost. We binarized the labels so that the letters before "N" is class 0 and the letters after and including "N" are class 1. We split the examples 12000/4000/4000 for training/validation/test sets. We train RBF SVM and RF (500 trees) with cross-validation as $f_0$. RBF SVM achieves the higher accuracy of 0.978 compared to RF 0.961.

To run the greedy algorithm, we first cross validate L1-regularized logistic regression with 20 C parameters in logspace of [1e-3,1e1]. For each C value, we obtain a classifier and we order the absolute values of its components and threshold them at different levels to recover all 16 possible supports (ranging from 1 feature to all 16 features). We save all such possible supports as we sweep C value. Then for each of the supports we have saved, we train a L2-regularized logistic regression only based on the support features with regularization set to 1 as $f_1$. The gating $g$ is then learned using L2-regularized logistic regression based on the same feature support and pseudo labels of $f_1$ - 1 if it is correctly classified and 0 otherwise. To get different cost-accuracy tradeoff, we sweep the class weights between 0 and 1 so as to influence $g$ to send different fractions of examples to the $f_0$.

To run ADAPT-LIN, we initialize $g$ to be 0 and $f_1$ to be the output of the L2-regularized logistic regression based on all the features. We then perform the alternative minimization for 50 iterations and sweep $\gamma$ between [1e-4,1e0] for 20 points and P$_{\text{full}}$ in [0.1,0.9] for 9 points.

To run ADAPT-GBRT, we use 500 depth 4 trees for $g$ and $f_1$ each. We initialize $g$ to be 0 and $f_1$ to be the GreedyMiser output of 500 trees. We then perform the alternative minimization for 30 iterations and sweep $\gamma$ between [1e-1,1e2] for 10 points in logspace and P$_{\text{full}}$ in [0.1,0.9] for 9 points. In addition, we also sweep the learning rate for GBRT for 9 points between [0.1,1].

For fair comparison, we run GREEDYMISER with 1000 depth 4 trees so that the model size matches that of ADAPT-GBRT. The learning rate is swept between [1e-5,1] with 20 points and the $\lambda$ is swept between [0.1, 100] with 20 points.

Finally, we evaluate all the resulting systems from the parameter sweeps of all the algorithms on validation data and choose the efficient frontier and use the corresponding settings to evaluate and plot the test performance.

## 0.6 MiniBooNE Particle Identification and Forest Covertype Datasets [4]:

The MiniBooNE data set is a binary classification task to distinguish electron neutrinos from muon neutrinos. There are 45523/19510/65031 examples in training/validation/test sets. Each example has 50 features, each with unit cost. The Forest data set contains cartographic variables to predict 7 forest cover types. There are 36603/15688/58101 examples in training/validation/test sets. Each example has 54 features, each with unit cost.

We use the unpruned RF of BUDGETPRUNE [6] as $f_0$ (40 trees for both datasets.) The settings for ADAPT-GBRT are the following. For MiniBooNE we use 100 depth 4 trees for $g$ and $f_1$ each. We initialize $g$ to be 0 and $f_1$ to be the GreedyMiser output of 100 trees. We then perform the alternative minimization for 50 iterations and sweep $\gamma$ between [1e-1,1e2] for 20 points in logspace and $P_{full}$ in [0.1,0.9] for 9 points. In addition, we also sweep the learning rate for GBRT for 9 points between [0.1,1]. For Forest we use 500 depth 4 trees for $g$ and $f_1$ each. We initialize $g$ to be 0 and $f_1$ to be the GreedyMiser output of 500 trees. We then perform the alternative minimization for 50 iterations and sweep $\gamma$ between [1e-1,1e2] for 20 points in logspace and $P_{full}$ in [0.1,0.9] for 9 points. In addition, we also sweep the learning rate for GBRT for 9 points between [0.1,1].

For fair comparison, we run GREEDYMISER with 200 depth 4 trees so that the model size matches that of ADAPT-GBRT for MiniBooNE. We run GREEDYMISER with 1000 depth 4 trees so that the model size matches that of ADAPT-GBRT for Forest.

Finally, we evaluate all the resulting systems from the parameter sweeps on validation data and choose the efficient frontier and use the corresponding settings to evaluate and plot the test performance.

## 0.7 Yahoo! Learning to Rank[2]:

This ranking dataset consists of 473134 web documents and 19944 queries. Each example is associated with features of a query-document pair together with the relevance rank of the document to the query. There are 519 such features in total; each is associated with an acquisition cost in the set {1,5,20,50,100,150,200}, which represents the units of CPU time required to extract the feature and is provided by a Yahoo! employee. The labels are binarized into relevant or not relevant. The task is to learn a model that takes a new query and its associated documents and produce a relevance ranking so that the relevant documents come on top, and to do this using as little feature cost as possible. The performance metric is Average Precision @ 5 following [6].

We use the unpruned RF of BUDGETPRUNE [6] as $f_0$ (140 trees for both datasets.) The settings for ADAPT-GBRT are the following. we use 100 depth 4 trees for $g$ and $f_1$ each. We initialize $g$ to be 0 and $f_1$ to be the GREEDYMISER output of 100 trees. We then perform the alternative minimization for 20 iterations and sweep $\gamma$ between [1e-1,1e3] for 30 points in logspace and $P_{full}$ in [0.1,0.9] for 9 points. In addition, we also sweep the learning rate for GBRT for 9 points between [0.1,1].

For fair comparison, we run GREEDYMISER with 200 depth 4 trees so that the model size matches that of ADAPT-GBRT for Yahoo.

Finally, we evaluate all the resulting systems from the parameter sweeps on validation data and choose the efficient frontier and use the corresponding settings to evaluate and plot the test performance.

## 0.8 CIFAR10 [5]:

CIFAR-10 data set consists of 32x32 colour images in 10 classes. 400 features for each image are extracted using technique described in [3]. The data are binarized by combining the first 5 classes into one class and the others into the second class. There are $19,761/8,468/10,000$ examples in training/validation/test sets. BUDGETPRUNE starts with a RF of 40 trees, which achieves an accuracy of 69%. We use an RBF-SVM as $f_0$ that achieves a test accuracy of 79.5%. The settings for ADAPT-GBRT are the following. we use 200 depth 5 trees for $g$ and $f_1$ each. We initialize $g$ to be 0 and $f_1$ to be the GREEDYMISER output of 200 trees. We then perform the alternative minimization for 50 iterations and sweep $\gamma$ between [1e-4,10] for 15 points in logspace and $P_{full}$ in [0.1,0.9] for 9 points. In addition, we also sweep the learning rate for GBRT for 10 points between [0.01,1].

For fair comparison, we run GREEDYMISER with 400 depth 5 trees so that the model size matches that of ADAPT-GBRT.

Finally, we evaluate all the resulting systems from the parameter sweeps on validation data and choose the efficient frontier and use the corresponding settings to evaluate and plot the test performance.