[Reviews · NeurIPS 2017]

Reviewer 1



The paper proposes a framework to train a composite classifier under a test-time budget for feature construction. The classifier consists of a high-performance model, built with no constraints on cost, a low-prediction cost model and a gating function which selects one of the two models to be used for a test sample. The setup of the problem is fairly specific: there are no constraints on complexity or feature computation cost during training time, however, at test time the prediction should be made as well as possible using as few features as possible (with a tradeoff). While several domain where this setup is relevant are mentioned in the intro, no details are given on the intended application until the experiment section. Also, only for one of the datasets on which this was tested (the Yahoo one) is runtime prediction even a consideration. The other datasets seem to have been arbitrarily selected. All these factors create the impression that the setup is somewhat artificial. The authors did a thorough job of reviewing budget-constrained classifiers, though they might also consider “Trading-Off Cost of Deployment Versus Accuracy in Learning Predictive Models” IJCAI ’16, by Daniel P. Robinson and Suchi Saria. Sections 3 and 4 (problem setup and algorithms) are well presented, the optimization is well designed and the I-projection solution for the linear version is clever. The one part that’s not entirely clear is Remark 1. “train a new pair of gating and prediction models to approximate the composite system” - why wouldn’t this result in the same solution? what changes in the re-training? wouldn’t you want to just approximate the expensive component in the prediction using its assigned samples? The experiments on synthetic data help in explaining why complex predictors can be summarized cost-efficiently with the current model and show some of the advantages of the method. However, maybe this part can be shortened and more room be allocated to a discussion on why this method outperforms GreedyMiser. What I find a surprising is that, in Figure 4, the performance of GreedyMiser (gradient-boosted trees) is nowhere near that of the other two models even when the cost is high. This might happen because the algorithm is improperly used or the base classifier is either badly tuned or in general not as well-performing as random forests. The classification cost differs per sample; do does the graph show the average cost for a point or the total cost across all test samples averaged over several runs? Typo: line 168: “g can effective” should be “g can effectively”. Update following the rebuttal: the authors have answered the questions and concerns about Figure 4, so the decision is updated accordingly.

Reviewer 2



1. Summary of paper This paper introduced an adaptive method for learning two functions. First, a `cheap' classifier that incurs a small resource cost by using a small set of features. Second, a gating function that decides whether an input is easy enough to be sent to the cheap classifier or is too difficult and must be sent to a complex, resource-intensive classifier. In effect, this method bears a lot of similarity to a 2-stage cascade, where the first stage and the gating function are learned jointly. 2. High level subjective The paper is written clearly for a reader familiar with budgeted learning, although I think a few parts of the paper are confusing (I'll go into this more below). I believe the idea of approximating an expensive, high-accuracy classifier is new (cascades are the most similar, but I believe cascades usually just consider using the same sort of classifiers in every stage). That said, it is very similar to a 2-stage cascade with different classifiers. I think there are certainly sufficient experiments demonstrating this method. Likely this method would be interesting to industry practitioners due to the increased generalization accuracy per cost. 3. High level technical Why not try to learn a cost-constrained q function directly? If you removed the KL term and used q in the feature cost function instead of g, wouldn't you learn a q that tried as best to be accurate given the cost constraints? Did you learn a g separately in order to avoid some extra non-convexity? Or is it otherwise crucial? I would recommend explaining this choice more clearly. I would have loved to see this model look less like a cascade. For instance, if there were multiple expensive 'experts' that were learned to be diverse w.r.t each other and then multiple gating functions along with a single cheap classifier were learned jointly, to send hard instances to one of the experts, then this would be a noticeable departure from a cascade. Right now I'm torn between liking this method and thinking it's a special cascade (using techniques from (Chen et al., 2012) and (Xu et al., 2012)). 4. Low level technical - Line 161/162: I don't buy this quick sentence. I think this is much more complicated. Sure we can constrain F and G to have really simple learners but this won't constrain computation per instance, and the only guarantee you'll have is that your classifier is at most as computationally expensive as the most expensive classifier in F or G. You can do better by introducing computational cost terms into your objective. - Line 177: There must be certain conditions required for alternating minimization to converge, maybe convexity and a decaying learning rate? I would just clarify that your problem satisfies these conditions. - I think Figure 5 is a bit confusing. I would move Adapt_Gbrt RBF to the top of the second column in the legend, and put GreedyMiser in its original place. This way for every row except the last we are comparing the same adaptive/baseline method. 5. Summary of review While this method achieves state-of-the-art accuracy in budgeted learning, it shares a lot of similarities with previous work (Chen et al., 2012) and (Xu et al., 2012) that ultimately harm its novelty.

Reviewer 3



The paper proposes to learn under budget constraints (of acquiring a feature) by jointly learning a low-cost prediction and gating model. Specifically, it gives two algorithms - for linear models and non-parametric models (gradient boosted trees). The paper is well presented explaining the motivations and the scope of the work. Concerns: 1. It would be interesting to make a further study about the gains achieved for each of the datasets. For example, CIFAR-10 offers little gains as compared MiniBooNE dataset. Is this because the data does not satisfy some of the assumptions made in the paper ? If so, what are the datasets (from a statistical sense) for which this technique will not have significant cost reduction ? 2. Figure 4 is squished. Please correct this add and gridlines to increase readability. 3. L50 - low-cost gating 4. It would be cleaner to avoid notational overload such as g(x) = g^Tx.